# Prompt rewetting of drained peatlands reduces climate warming despite methane emissions

Anke Günther [1⊠], Alexandra Barthelmes[2,3], Vytas Huth [1], Hans Joosten [2,3], Gerald Jurasinski [1], Franziska Koebsch[1] & John Couwenberg[2,3]

Peatlands are strategic areas for climate change mitigation because of their matchless carbon stocks. Drained peatlands release this carbon to the atmosphere as carbon dioxide ($CO_2$). Peatland rewetting effectively stops these $CO_2$ emissions, but also re-establishes the emission of methane ($CH_4$). Essentially, management must choose between $CO_2$ emissions from drained, or $CH_4$ emissions from rewetted, peatland. This choice must consider radiative effects and atmospheric lifetimes of both gases, with $CO_2$ being a weak but persistent, and $CH_4$ a strong but short-lived, greenhouse gas. The resulting climatic effects are, thus, strongly time-dependent. We used a radiative forcing model to compare forcing dynamics of global scenarios for future peatland management using areal data from the Global Peatland Database. Our results show that $CH_4$ radiative forcing does not undermine the climate change mitigation potential of peatland rewetting. Instead, postponing rewetting increases the long-term warming effect through continued $CO_2$ emissions.

[1] University of Rostock, Faculty of Agricultural and Environmental Studies, Landscape Ecology, Rostock, Germany. [2] University of Greifswald, Faculty of Mathematics and Natural Sciences, Peatland Studies and Paleoecology, Greifswald, Germany. [3] Greifswald Mire Centre (GMC), Greifswald, Germany. ⊠email: anke.guenther@uni-rostock.de

Each year, drained peatlands worldwide emit ~2 Gt carbon dioxide ($CO_2$) by microbial peat oxidation or peat fires, causing ~5% of all anthropogenic greenhouse gas (GHG) emissions on only 0.3% of the global land surface[1]. A recent study states that the effect of emissions from drained peatlands in the period 2020–2100 may comprise 12–41% of the remaining GHG emission budget for keeping global warming below +1.5 to +2 °C[2]. Peatland rewetting has been identified as a cost-effective measure to curb emissions[3], but re-establishes the emission of methane ($CH_4$). In light of the strong and not yet completely understood impact of $CH_4$ on global warming[4,5] it may seem imprudent to knowingly create or restore an additional source. Furthermore, there is considerable uncertainty on emissions from rewetted peatlands and some studies have reported elevated emissions of $CH_4$ compared with pristine peatlands[6–9].

The trade-off between $CH_4$ emissions with and $CO_2$ emissions without rewetting is, however, not straightforward: $CH_4$ has a much larger radiative efficiency than $CO_2$[10]. Yet, the huge differences in atmospheric lifetime lead to strongly time-dependent climatic effects. Radiative forcing of long-term GHGs (in case of peatlands: $CO_2$ and $N_2O$) is determined by cumulative emissions, because they factually accumulate in the atmosphere. In contrast, radiative forcing of near-term climate forcers (in case of peatlands: $CH_4$) depends on the contemporary emission rate multiplied with the atmospheric lifetime[10,11], because resulting atmospheric concentrations quickly reach a steady state of (sustained) emission and decay. Meanwhile, common metrics like global warming potential (GWP) and its sustained flux variants[11,12] fail to account for temporal forcing dynamics. These different atmospheric dynamics are relevant for the question how the various management scenarios will influence global climate and whether a scenario will amplify or attenuate peak global warming, i.e., the maximum deviation in global surface temperatures relative to pre-industrial times. An amplification of peak warming increases the risk of reaching major tipping points in the Earth's climate system[13,14].

Here, we explore how the different lifetimes of $CO_2$/$N_2O$ vs. $CH_4$ play out when assessing options for peatland rewetting as a climate warming mitigation practice by comparing five global scenarios (Table 1). These scenarios represent extreme management options and exemplify the differences caused by timing and extent of rewetting. For our modeling exercise, we focus on the direct human-induced climatic effects and conservatively assume pristine peatlands to be climate-neutral. Further, we assume that the maximum peatland area to be drained during the 21st century equals the area that is already drained in 2018 (505,680 km², Global Peatland Database[15]) plus an additional ~5000 km² per year (average net increase of drained peatland area between 1990 and 2017[16]). For all scenarios, we apply IPCC default emissions factors[17] as sustained fluxes. To compare the radiative forcing

effects of the different GHGs, we use a simplified atmospheric perturbation model that has been shown to provide reliable estimates of the climatic effects of peatlands[18] (see Methods). Our results show that total radiative forcing quickly reaches a plateau after rewetting, because of the halted emissions of $CO_2$/$N_2O$ of rewetted peatlands and the short atmospheric lifetime of any emitted $CH_4$. In contrast, postponing rewetting has a long-term warming effect resulting from continued $CO_2$ emissions. Warnings against $CH_4$ emissions from rewetted peatlands are therefore unjustified in the context of effective climate change mitigation.

## Results and Discussion

**Radiative forcing dynamics of global scenarios.** Rewetting of drained peatlands instantly leads to climatic benefits compared with keeping the status quo (Fig. 1). In case of rewetting all drained peatlands (scenarios Rewet_All_Now and Rewet_All_Later, see Table 1) the radiative forcing stops increasing followed by a slow decrease. Since the response of global

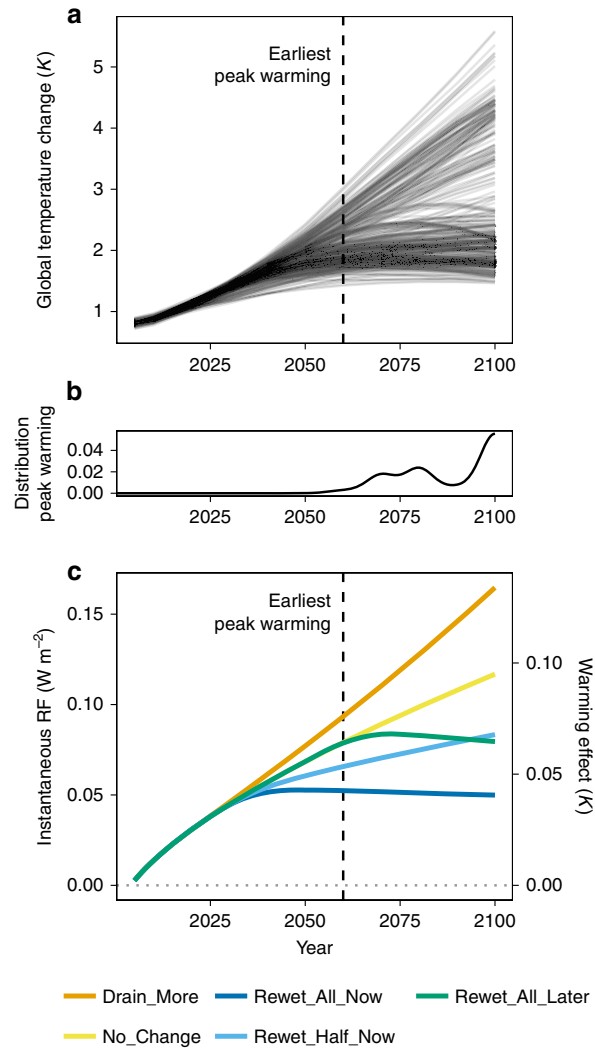

**Fig. 1 Global warming and climatic effects of peatland management.** Mean global temperature change relative to 2005 (**a**) and frequency distribution of the timing of peak warming (**b**) according to AR5 model pathways (downloaded from IAMC AR5 Scenario Database) are shown compared with radiative forcings (RF) and estimated instantaneous warming effects of global peatland management scenarios (panel **c**, own calculations). Please note that in panel **c**) forcing of peatlands that remain pristine is assumed to be zero.

**Table 1 Global scenarios of peatland management.**

| Scenario | Description |
|---|---|
| Drain_More | The area of drained peatland continues to increase from 2020 to 2100 at the same rate as between 1990 and 2017 |
| No_Change | The area of drained peatland remains at the 2018 level |
| Rewet_All_Now | All drained peatlands are rewetted in the period 2020–2040 |
| Rewet_Half_Now | Half of all drained peatlands are rewetted in the period 2020–2040 |
| Rewet_All_Later | All drained peatlands are rewetted in the period 2050–2070 |

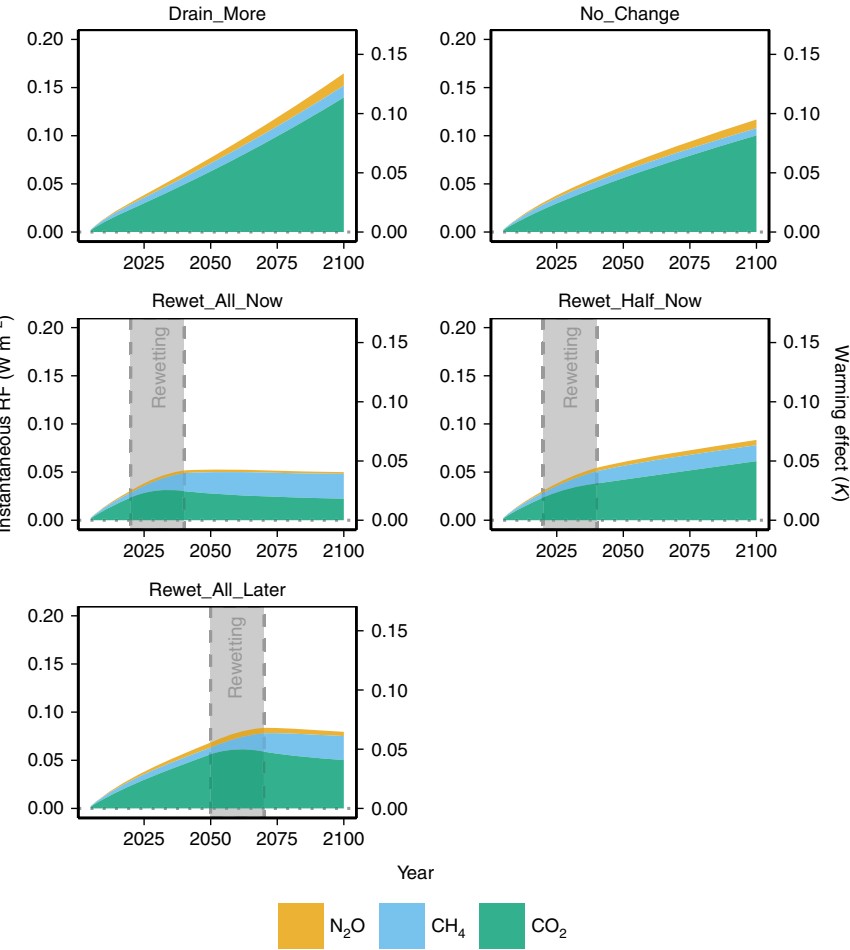

**Fig. 2 Climatic effects of peatland scenarios by greenhouse gas.** Contributions of the different greenhouse gases (nitrous oxide, $N_2O$, methane, $CH_4$, and carbon dioxide, $CO_2$) to total radiative forcing (RF) are shown with estimated warming effects in the modeled scenarios. The gray area shows the period of rewetting. Note that in the figure forcing of peatlands that remain pristine is assumed to be zero.

temperature is lagging behind changes in total radiative forcing by 15–20 years[19], peatlands should be rewetted as soon as possible to have most beneficial (cooling) effects during peak warming, which AR5 climate models expect to occur after ~2060 with increasing probability towards the end of the century (Fig. 1).

The overall climatic effect of peatland rewetting is indeed strongly determined by the radiative forcing of sustained $CH_4$ emissions (Fig. 2). However, because of the negligible or even negative emissions of $CO_2/N_2O$ of rewetted peatlands and the short atmospheric lifetime of $CH_4$, the total anthropogenic radiative forcing of all three GHGs combined quickly reaches a plateau after rewetting. Meanwhile, differences in radiative forcing between drainage (increased forcing) and rewetting scenarios (stable forcing) are mainly determined by differences in the forcing of $CO_2$ (Fig. 2). Rewetting only half of the currently drained peatlands (Rewetting_Half_Now) is not sufficient to stabilize radiative forcing. Instead, $CO_2$ from not-rewetted peatland keeps accumulating in the atmosphere and warming the climate. Note that in the Rewet_Half_Now scenario $CH_4$ forcing is more than half that of the Rewet_All_… scenarios, because drained peatlands also emit $CH_4$, most notably from drainage ditches. Comparing the scenarios Rewet_All_Now and Rewet_All_Later shows that timing of peatland rewetting is not only important in relation to peak temperature, but also with respect to the total accumulated $CO_2$ and $N_2O$ emissions in the atmosphere and the resulting radiative forcing (Fig. 2). These

patterns are valid also when considering possible future changes of new drainage rate or emission factors (Fig. 3).

**General conclusions for global peatland management.** Our simulations highlight three general conclusions: First, the baseline or reference against which peatland rewetting has to be assessed is the drained state with its large $CO_2$ emissions. For this reason, rewetted peatlands that are found to emit more $CH_4$ than pristine ones[9] are no argument against rewetting. Moreover, whereas rewetted peatlands may again become $CO_2$ sinks, the faster and larger climatic benefits of peatland rewetting result from the avoidance of $CO_2$ emissions from drained peatlands. Second, the climate effect is strongly dependent on the concrete point in time that rewetting is implemented. This fact is hitherto insufficiently recognized because it remains hidden by the common use of metrics that involve predetermined time horizons (like GWP or sustained flux variants of GWP). Finally, in order to reach climate-neutrality in 2050 (as implied by the Paris Agreement on limiting the increase in global temperature to well below 2 °C), it is insufficient to focus rewetting efforts on selected peatlands only: to reach the Paris goal, $CO_2$ emissions from (almost) all drained peatlands have to be stopped by rewetting[2].

Limiting global warming requires immediate reduction of global GHG emissions. It has been suggested that the negative climate effects of drained peatlands could be offset by growing highly-productive bioenergy crops[20] or wood biomass[21] as

substitute for fossil fuels. In this study, we did not include this option because similar biomass-based substitution benefits can also be reached by cultivating biomass on rewetted peatlands[22], i.e., without $CO_2$ emissions from drained peat soil.

In conclusion, without rewetting the world's drained peatlands will continue to emit $CO_2$, with direct negative effects on the magnitude and timing of global warming. These effects include a higher risk of reaching tipping points in the global climate system and possible cascading effects[13]. In contrast, we show that peatland rewetting can be one important measure to reduce climate change and attenuate peak global warming: The sooner drained peatlands are rewetted, the better it is for the climate. Although the $CH_4$ cost of rewetting may temporarily be substantial, the $CO_2$ cost of inaction will be much higher.

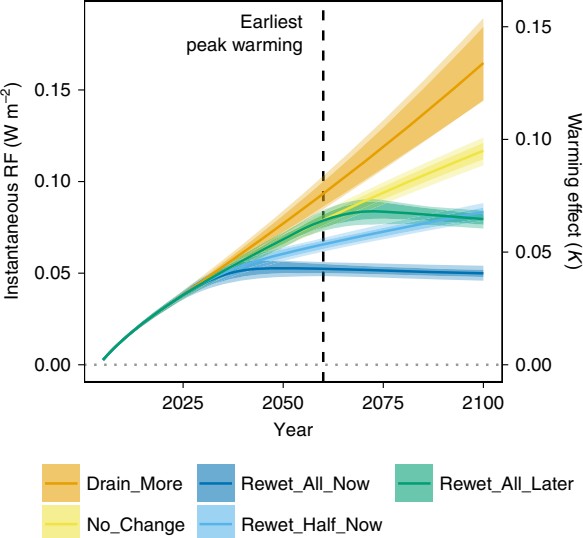

**Fig. 3 Modeling sensitivity to variation of input values.** The influence of modeling choices and uncertainty of emission factors on radiative forcing (RF) and on estimated warming effects is shown for the five global peatland scenarios. Error ranges represent the range (minimum to maximum) of radiative forcing resulting from random variations in ongoing drainage rate (1000–8000 km² per year) and emission factors (10 and 20% uncertainty of emission factor, represented by shading intensity).

## Methods

**Scenarios**. Drained peatland area was taken from the Global Peatland Database (GPD)[15], which includes, among other things, national data from the most recent UNFCCC National Inventory Submissions and Nationally Determined Contributions. We used data separated by IPCC climate zone (boreal, temperate, and tropical) and assigned land use categories. Available land use categories were Forest, Cropland, Deep-drained grassland, Shallow-drained grassland, Agriculture (i.e., either grassland or cropland when the original data source did not differentiate between these two categories), and Peat extraction (see Table 2). Because of their only small area and uncertain emission factors, arctic drained peatlands (~100 kha) were neglected. The average net increase of drained peatland area between 1990 and 2017[16] assumed for the Drain_More scenario includes the disappearance of drained peatlands that have lost all their peat deposits. Newly drained/rewetted area in the scenarios is distributed across the climatic zones (and land use classes) according to the relative proportions of today's drained peatland area. As future drainage—similar to the past two decades[16]—will probably focus on tropical and subtropical peatlands, our Drain_More scenario likely underestimates the climate effects of future drainage. For information on how variations in the assumed drainage rate and uncertainty of emission factors affected the displayed radiative forcing effects of the scenarios please see Fig. 3.

**Emissions**. Emission factors for each climate zone and land use category were taken from the IPCC Wetland supplement[17] that currently presents the most robust and complete meta-study of published emission data. We applied all emission factors as sustained fluxes. Emission factors were averaged for IPCC categories that were given at a higher level of detail (e.g., nutrient-poor vs. nutrient-rich boreal forest) than the available land use categories from the GPD. Equally, we averaged the supplied emission factors for grassland and cropland in order to obtain emission factors of the land use class Agriculture (see Table 2 for final aggregated emission factors and Supplementary Table 1 for exact aggregation steps). We included emissions from ditches and DOC exports by using emission factors and default cover fraction of ditches given by the IPCC[17] (Supplementary Table 1). Since the IPCC Wetlands Supplement does not provide an emission factor for $CH_4$ from tropical peat extraction sites, we assumed the same $CH_4$ emissions as for temperate/boreal peat extraction. Values of the emission factors could change slightly when more emission data becomes available. To cover this possibility, we randomly varied all emission factors within a range of 10–20% uncertainty in our sensitivity analysis (Fig. 3). Please note that these uncertainty ranges do not correspond to the confidence intervals given by the IPCC Wetlands Supplement, which describe the observed variability of emissions from individual peatlands. Since our analyses take a global perspective, our sensitivity analyses instead cover possible changes of the mean emissions (i.e., emission factors). In addition, individual studies have discussed the presence of a $CH_4$ peak for the first years after rewetting[7,8]. Although this is likely not a global phenomenon[23], please see Supplementary Fig. 1 for an estimate of the uncertainty related to possible $CH_4$ peaks.

**Radiative forcing**. The forcing model uses simple impulse-response functions[24] to estimate radiative forcing effects of atmospheric perturbations of $CO_2$, $CH_4$, and $N_2O$ fluxes[12]. Perturbations of $CH_4$ and $N_2O$ were modeled as simple exponential decays, while $CO_2$ equilibrates with a total of five different pools at differing speeds. For $CO_2$, we adopted the flux fractions and perturbation lifetimes used by ref. [18]. In

**Table 2 Areas of drained peatland (kha) by climate zone and land use category according to the Global Peatland Database, together with aggregated emission factors.**

| Climatic zone | Land use category | Area (kha) | $CO_2$ (t ha⁻¹ a⁻¹) | $CH_4$ (kg ha⁻¹ a⁻¹) | $N_2O$ (kg ha⁻¹ a⁻¹) |
|---|---|---|---|---|---|
| Boreal | Forest | 5474 | 2.5 | 9.8 | 2.6 |
| | Cropland | 262 | 27.9 | 58.3 | 19.4 |
| | Deep-drained grassland | 426 | 20.2 | 59.6 | 14.2 |
| | Shallow-drained grassland | 0 | — | — | — |
| | Agriculture | 3420 | 24.1 | 43.0 | 16.8 |
| | Peat extraction | 333 | 10.2 | 32.9 | 0.5 |
| | Rewetted | — | −1.3 | 123.6 | 0 |
| Temperate | Forest | 6315 | 10.3 | 7.9 | 4.3 |
| | Cropland | 2528 | 28.6 | 58.3 | 19.4 |
| | Deep-drained grassland | 3405 | 22.3 | 73.5 | 12.3 |
| | Shallow-drained grassland | 2422 | 13.6 | 63.4 | 2.4 |
| | Agriculture | 8389 | 21.0 | 55.8 | 10.1 |
| | Peat extraction | 662 | 10.8 | 32.9 | 0.5 |
| | Rewetted | — | −0.4 | 205.9 | 0 |
| Tropical | Forest | 7235 | 22.0 | 50.0 | 3.7 |
| | Cropland | 305 | 45.0 | 118.9 | 4.2 |
| | Deep-drained grassland | 70 | 37.4 | 52.0 | 7.7 |
| | Shallow-drained grassland | 0 | — | — | — |
| | Agriculture | 9314 | 42.5 | 96.6 | 5.4 |
| | Peat extraction | 8 | 10.1 | 32.9 | 5.6 |
| | Rewetted | — | 1.9 | 166.5 | 0 |

Emission factors assumed for rewetted peatlands are also shown for each climatic zone.

the model, we assume a perfectly mixed atmosphere without any feedback mechanisms but include indirect effects of $CH_4$ on other reagents[10].

Climatic effects of $CO_2$ from $CH_4$ oxidation should not be considered for $CH_4$ from biogenic sources[10]. However, although the large majority of $CH_4$ from peatlands stems from recent plant material (a biogenic source), the proportion of fossil $CH_4$ (from old peat) may be substantial in some cases[25]. Thus, we conservatively included the climatic effect of $CO_2$ from $CH_4$ oxidation in our analyses. Overall, this forcing comprised only 5–7% of the $CH_4$ radiative forcing and only ~1–3% of total radiative forcing.

We compare the radiative forcing trajectories of the various peatland management scenarios with the global temperature change as projected by all available pathways of IPCC's AR5 and use the same starting year 2005 as these pathways. Further, we estimated the approximate effects of radiative forcing on global mean temperature as ~1 K per 1.23 W/m² radiative forcing[26].

## Data availability
The models for projected temperature change were downloaded from IAMC AR5 Scenario Database (available at https://secure.iiasa.ac.at/web-apps/ene/AR5DB). Emission factors and peatland cover data are entirely included in the paper.

## Code availability
The code for the atmospheric perturbation model can be found in the supplementary information. Code was written by A. Günther in the R programming language.

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

## Acknowledgements
The European Social Fund (ESF) and the Ministry of Education, Science and Culture of Mecklenburg-Western Pomerania funded this work within the scope of the project WETSCAPES (ESF/14-BM-A55-0030/16 and ESF/14-BM-A55-0031/16). G.J. received funding within the framework of the Research Training Group Baltic TRANSCOAST from the DFG (Deutsche Forschungsgemeinschaft) under grant number GRK 2000 (www.baltic-transcoast.uni-rostock.de). This is Baltic TRANSCOAST publication no. GRK2000/0032. V.H. gratefully acknowledges funding by the Federal Agency of Nature Conservation (BfN, grant number: 3516892003) and by the European Regional Development Fund (ERDF) distributed through the NBank.

## Author contributions
A.G., J.C., G.J., F.K., and V.H. conceived the study. A.G., A.B., J.C., and H.J. assembled input data. A.G. implemented the simulation model with contributions from J.C. All authors discussed the results and implications. A.G. led writing of the manuscript with comments/edits from all authors.

## Competing interests
The authors declare no competing interests.
