## [Peer Review File · Nature Communications]

Reviewers' comments:

Reviewer #1 (Remarks to the Author):

Gunther et al. Prompt rewetting of drained peatlands reduces climate warming despite methane emissions

In this communication paper Gunther et al. show the radiative forcing trajectory for several peatland rewetting scenarios. They conclude that rewetting needs to take place before 2050 to avoid overlapping with the peak warming.

I find the topic highly relevant and timely. There is not much literature addressing the climate impacts of rewetting, or how the goals and timeline of rewetting impact the goals of halting the global warming to 1.5 or 2 degrees. In my understanding, the paper is novel and of high importance for the climate change community and the society. While I find the topic very fascinating, I have several issues which need to be addressed before the paper can be accepted.

First of all, the description of methods and assumptions is very brief and too vague. Please find several comments about that below. Second, I recommend that the RF calculations will be done by looking at the differences between the emissions/sinks from rewetting vs. a reference (the situation as it is now). This will also make the interpretation of the RF curves easier. Third, I think that some discussion of the geographical differences is needed, addressing the question where the rewetting would be most efficient. I think it is a bit problematic that peatlands of the world have been lumped together. Would it be more effective in some region as compared to another region? Now the conclusion is that all peatlands should be rewetted, the quicker the better. Does it mean then that if we cannot rewet all, we should not rewet any?

It is also clear that not all rewetted wetlands have similar CH₄ emissions or CO₂ uptake after the rewetting. If there are wetland types which produce high CH₄ emission but do not turn into high sinks, would it be wise to avoid rewetting some of these? Finally, how well we actually know the emissions of rewetted peatlands to generalize them? How sensitive are the calculations for these uncertainties, and more specifically to the initial peak? This kind of discussion/analysis is totally missing. My last point is that there is no discussion on the issue of avoiding causing the emissions during the "peak warming". Why it is so important?

Based on these points raised here I am suggesting that the MS needs major revisions to be accepted. I strongly encourage the authors to move some of these missing information into the supplement, if there is not enough space in the main text.

Detailed comments:

line 27: how well these peaks are understood? Do they occur in all types of peatlands? The way the text is written gives a feeling of a lots of uncertainties (possible, may be associated...) even though the peaks have a central role here.

line 33: may be associated? Isn't this clear? How is this uncertainty reflected in the RF calculations?

lines 50-51: In the rewetting scenarios the reference situation is the emissions from drained peatlands. However, in the "Drain_more" scenario the drained situation cannot be used as a reference. In that sense the scenarios are not balanced. The reference scenario for the drain_more should be given.

line 60: is this valid for "drain_more" only?

line 62: I am missing justification for the strong spike in the beginning. Why 10 times the natural?

How sensitive your calculation is for this number? What if the size of the spike was 100 times the natural emissions, or less than 10? Would it change your conclusions? Are the results of the only two papers generalizable? Now there is a sensitivity test for the areas of drained/rewettered peatlands, but not for the flux rates, which I suppose are much more critical than the area. Also, where is the peak duration of 0-5 years stemming from?

lines 66-67: Here status quo means that the reference is "No_change", but it is not explicitly stated. It would be better to present all the results as a difference between the scenario and the reference. Then, also the 'warming overshoot' would be more easily and clearly detected.

I am missing discussion related to the selection and use of climate scenarios. This has not been explained at all and should be done either in the main text or in supplement. Why are you using exactly these scenarios? Why not the 1.5C scenarios which are more recent and show an earlier peak warming? Why peak warming is so important? And which one is more important: the quick warming impact, or the cooling afterwards?

If the RF values in Fig. 1 were shown as cumulative values one could easily see when the warming due to increased CH₄ emissions have been compensated. Also, I find no discussion of the dynamics of the RF curve. Why is the curve for "drain_all" saturating so quickly? Because there will be more CO₂ in the atmosphere and any addition/removal will have smaller relative impact? These explanations and discussion should be provided somewhere.

I am also missing more detailed discussion of the RF dynamics and the warming effect: Questions related to the RF calculation which should be addressed somewhere: Why is the temperature change looked at since 2005? Why exactly this year? You are saying that during the 21st century, peatland drainage will warm the climate by 0.2 K. This is 20% of the already observed global warming. How realistic it is?

lines 73-74: the emissions of rewettered peatlands have not been shown anywhere. Please provide them in a table similarly to the emissions of the drained peatlands. Also, N₂O is first time mentioned here. Some description of the processes which lead to the suppressed emissions is necessary.

line 75: "...quickly reaches a plateau..." But what happens if you reduce the climate impact of the reference here?

lines 75-76: "differences in radiative forcing between the scenarios are mainly determined by CO₂": this is not at all clear from the figure 2. Again, would be more informative to look at the differences between the emissions from /sinks of reference vs. rewettered.

In Fig. 1c you are stating that it does not include forcing of peatlands that remain pristine. If this means that emissions/sinks of pristine mires have not been used as a reference for the "drain_more" scenario, it is not OK. For the "drain_more" scenario it is exactly the fluxes of pristine mire which should be used as a reference.

lines 80-83: Again, would be good to discuss why is the "peak temperature" so important here?

line 82: "total long-term forcing" what is this? Definition?

line 91: "stabilize global climate" is vague terminology. What does stabilization of climate mean?

lines 94-98: I do not really understand this paragraph. How representative are the references used here? At least the other one (Minkinen et al.) is not really a productive forest, but a poorly growing one. I do not understand this justification: "In this study, we did not include this option because wet cultivation methods ('paludiculture') could provide similar substitution benefits without CO₂ emissions

from drained peat soil.”

line 101: why “if”? Do we expect or do we not? This was the main assumption in this MS. Are you not sure about it?

line 102: why it is important to avoid the peak warming?

lines 112-113: what are the consequences of this assumption? How sensitive your result is to this?

Table M1: fluxes from the rewetted mires are missing

lines 120-121: some critical discussion of the IPCC emission factors would be welcomed. What happens with the averaging what you are doing? How reliable are the estimates for the ditch emissions? How realistic it is that the amount of ditches in agricultural peatlands is 5%, even though the number of underground drains seems to be relatively high? The emissions of ditches and DOC export are quite critical for the whole exercise, as they turn the drained peat soils in many regions into sources. Some discussion on that topic would be very important.

Reviewer #2 (Remarks to the Author):

Prompt rewetting of drained peatlands reduces climate warming despite methane emissions Anke Günther¹, Alexandra Barthelmes^{2,3}, Vytas Huth¹, Hans Joosten^{2,3}, Gerald Jurasinski¹, Franziska Koebisch¹, John Couwenberg^{2,3}

This is an interesting and nuanced paper that is needed on this topic relating to the unintended consequences of rewetting peatlands to sequester carbon. Many of us have learned that there is no free lunch and the process of flooding, which turns off aerobic respiration to favor long term carbon sequestration, also produced conditions that promote methane emissions. Methane is a strong greenhouse gas and with wetlands, the fluxes of methane are very, very strong. My personal concern working on this the topic is the impression some may make that we should not restore wetlands if the methane fluxes are so large. Hence it is warranted to provide a careful analysis to inform policy makers to prevent them from ‘throwing the baby out with the bath water’. Methane is very reactive with OH radical so its lifetime is only about 15 years, compared to CO₂, with a 300 year plus lifetime. So even if methane is produced it will not linger in the atmosphere indefinitely, though it will be converted to CO₂..that is a different story.

I also add that studies like this are important because they are needed to lend nuance to using natural solution to climate change. My one lingering worry is that some policy maker may conclude that restoring wetlands is bad because they also produce methane a stronger greenhouse gas. But we must not throw the proverbial baby out with the bath water. Ecological restoration has many co benefits that need to be considered and weighed, too. Including permanence of the C sink (most upland sinks have a relative short residence time compared to peatlands), protection of land due to sea level rise and storm surges, filtering water, habitat for birds and fish.

Overall, I firmly agree more with the central and salient conclusion of the authors

..., without rewetting the world’s drained peatlands will continue to emit CO₂, with direct effects on the magnitude and timing of peak global warming. These CO₂ emissions can effectively be stopped by rewetting. Especially if we expect large CH₄ emission spikes upon rewetting, we should rewet as soon as possible, so that these CH₄ emissions contribute as little as possible to peak warming. Although the CH₄ cost of rewetting may temporarily be substantial, the CO₂ cost of inaction will be much higher.

I like the aspect of the study that raises the question about restoring wetlands now, so the methane effect gets washed out soon enough, as compared to restoring wetlands later. It all makes me think about historical formation of wetlands, preindustrial. If the methane effect did not wash out we may

have experiences a bit more warming than we have between glacial and interglacial periods.

Methods

This paper addresses a so what question in scale by using the global peatland database, rather than focus on a few sites as some of us are guilty of. This is needed to address magnitude and impact at the scale of the atmosphere that processes all of this; I stress this because my one worry with peatland is that they are a super strong source but a small area, so we need to address these issues head on as this paper does.

This work lets the authors show that CH₄ radiative forcing does not undermine the climate change mitigation potential of peatland rewetting. Instead, postponing rewetting increases the long-term warming effect of continued CO₂ emissions. Unlike CO₂ (and N₂O) from drained peatlands that accumulates in the atmosphere, possible CH₄ emission spikes upon rewetting do not add to expected peak warming when rewetting occurs before 2050

Regarding to the computations of greenhouse warming potential I want to make sure they are using the newer Neubauer and Megonigal 2015 Ecosystems Moving beyond global warming potential, .. sustained warming potential method, rather than the older and simpler pulse method. It seems they are using the pulse method. For the sake of doing science best, and in its most defensible manner I strongly urge them to use the better methods.

One of the other reasons I like this work is it helps support wetland development over upland development. The later are highly sold, but their residence time remains relatively short. I find undecomposed corms in our peatlands that have been aged 4000 years! Putting carbon in trees or soil of ag or grasslands is bound to return to the atmosphere in less than 100 years in most locations. My conclusion, is to publish after revision and recompute the analysis with the sustained greenhouse warming potential method. Methane is not emitted as a pulse after rewetting drained lands. It is sustained for many years.

Reviewer #3 (Remarks to the Author):

Review of MS 'Prompt rewetting of drained peatlands reduces climate warming despite methane emissions' by Günther et al.

In the manuscript, authors calculate radiative forcing associated with different options for the future management of peatlands and pay particular attention to the seemingly controversial effect of methane. This is an important and timely piece of work although the main finding, namely that despite high methane release peatland rewetting is beneficial with respect to the overall forcing as compared to business as usual is not novel. The different life times and corresponding consequences for calculating radiative forcing over longer time scales has been widely discussed in the literature. The MS merits publication in Nature Communications but authors should pay attention to my major concerns as listed below.

It is not clear whether the exhaustion of peat deposits over time, after drainage, is considered in the calculated scenarios until 2100. Loosing part of former organic soils with drainage will reduce peatland derived CO₂ emissions over time.

Figure 1 provides an overview of scenario outcomes. Panel 'a' obviously shows various calculations but it remains unclear what they do represent.

In line 88, authors underpin a fact which has long been known (e.g., Frohling and Roulet 2008). There is less novelty in this finding than stated, although the consideration for different scenarios of future peatland use is appreciated.

In line 91, authors make a political claim that is, in my view, not appropriate for a scientific journal.

There is particular need to provide much better explanation of methods:

Authors are requested to explicitly explain i) how the climate zones are defined, ii) where the land-use and management information (deep vs. shallow drained grassland) is coming from.

Table M1 gives an overview of land use classes and assigned EF's of drained but not of rewetted peatlands. The latter should be included.

Table M1 and line 124. It is unclear what land use class 'agriculture' refers to given that cropland and grassland, which make up agriculture as a whole, are already listed. What does 'agriculture' include and how was this class delineated in the global data base?

Line 126: Authors are requested to write out all categories that were lumped and how.

In line 126-127 it is stated that EF's for tropical extraction sites equal those of boreal and temperate regions (owing to lack of data), but numbers in Table M1 do deviate from each other with the EF's from tropical being the lowest.

More details on the radiative forcing model are needed. From the description it seems that authors assume an instantaneous radiative forcing, but see Neubauer 2014 for a comparison between instantaneous and cumulative radiative forcing. Further, in line 62 it is mentioned that an initial strong CH₄ spike was considered. Does this correspond to the IPCC EF's for rewetted peatlands? Details on the origin and size of that spike are missing. The longevity of the CH₄ spike also remains unclear as well as what authors refer to as 'natural emissions'.

References

- Frolking S, Roulet NT. Holocene radiative forcing impact of northern peatland carbon accumulation and methane emissions. *Global Change Biology* 2007, 13(5): 1079-1088.
- Neubauer SC. On the challenges of modeling the net radiative forcing of wetlands: reconsidering Mitsch et al. 2013. *Landscape Ecology* 2014, 29(4): 571-577.

Reviewers' comments:

Reviewer #1 (Remarks to the Author):

In this communication paper Gunther et al. show the radiative forcing trajectory for several peatland rewetting scenarios. They conclude that rewetting needs to take place before 2050 to avoid overlapping with the peak warming.

I find the topic highly relevant and timely. There is not much literature addressing the climate impacts of rewetting, or how the goals and timeline of rewetting impact the goals of halting the global warming to 1.5 or 2 degrees. In my understanding, the paper is novel and of high importance for the climate change community and the society. While I find the topic very fascinating, I have several issues which need to be addressed before the paper can be accepted.

First of all, the description of methods and assumptions is very brief and too vague. Please find several comments about that below.

>> We have now included much more information about methods and assumptions. Please see our detailed answers below.

Second, I recommend that the RF calculations will be done by looking at the differences between the emissions/sinks from rewetting vs. a reference (the situation as it is now). This will also make the interpretation of the RF curves easier.

>> In our manuscript we actually do express all scenarios against the same reference, i.e. the situation in the year 2005. As customary with atmospheric perturbation models, the starting point is chosen arbitrarily (Dommain et al. 2018). We chose the year 2005 and expressed all management scenarios relative to that date, because it is the base year for most emission projections and the projected global temperature change in IPCC AR5. Please also see our comments below.

Third, I think that some discussion of the geographical differences is needed, addressing the question where the rewetting would be most efficient. I think it is a bit problematic that peatlands of the world have been lumped together. Would it be more effective in some region as compared to another region?

>> Firstly, the Paris Agreement implies that emissions from *all* drained peatlands have to reach net zero in 2050, so rewetting 'efficiency' would merely be a function of temporal prioritization within a limited time-frame.

Secondly, rewetting is most *climate*-effective where emissions from drained peatlands are highest, i.e. in the tropics. Whether rewetting there is also most *cost*-effective and feasible is, however, disputable, and depends on whether you follow business management or economic considerations,

choose a short- or long-term approach (with different options for discounting future costs of interventions and damage), implement post-rewetting land-use or not (Wichtmann et al. 2016), include political and ethical considerations (Bonn et al. 2016), etc. Including a discussion on all of the above (regionally differentiated) factors it would immensely increase the number of assumptions and the length of the manuscript and the many details and assumptions would erode the robust message of our result (again, a result that is entirely valid in light of the Paris Agreement). Therefore, we refrained from giving such recommendations, but include the sentence (L101 ff.):

“In order to reach climate-neutrality in 2050 as implied by the Paris Agreement, it is insufficient to focus rewetting efforts on selected peatlands only: to reach the Paris goal, CO₂ emissions from (almost) all drained peatlands have to be stopped by rewetting².”

Now the conclusion is that all peatlands should be rewetted, the quicker the better. Does it mean then that if we cannot rewet all, we should not rewet any?

>> Every ton of avoided carbon dioxide emission is better than none, but incomplete rewetting (i.e. every hectare of peatland remaining drained) keeps increasing the climate burden, as shown exemplarily by the curve for ‘Rewet_Half_Now’ in Figure 1.

It is also clear that not all rewetted wetlands have similar CH₄ emissions or CO₂ uptake after the rewetting. If there are wetland types which produce high CH₄ emission but do not turn into high sinks, would it be wise to avoid rewetting some of these?

>> It is a common misunderstanding that the climate effect of peatland rewetting is related to establishing a carbon *sink*. In fact, the main benefits result from stopping a CO₂ *source* (the drained peatland with its huge emissions). Therefore, peatlands that are rewetted and where CO₂ emissions have stopped, but that do not turn into CO₂ sinks will still be beneficial for the climate compared to keeping them drained, even when they emit CH₄. The effect would be different in case of persistent extreme CH₄ emissions, but these occur only very exceptionally, if at all, as literature shows. In such case, it is better to reduce these high CH₄ emissions (e.g. by careful water management, removing fresh biomass and nutrients through harvest or topsoil removal prior to rewetting, establishment of species with slowly degradable litter, ...) rather than keeping the peatland drained, because any emitted CH₄ will disappear much more quickly from the atmosphere (and stop warming the climate) than the emitted CO₂.

To clarify the difference between creating carbon sinks and avoiding CO₂ emissions, we now included a sentence in the discussion (L94-96). It reads:

“Moreover, whereas rewetted peatlands may again become CO₂ sinks, the faster and larger climatic benefits of peatland rewetting result from the avoidance of CO₂ emissions from drained peatlands.”

Finally, how well we actually know the emissions of rewetted peatlands to generalize them? How sensitive are the calculations for these uncertainties, and more specifically to the initial peak? This kind of discussion/analysis is totally missing.

>> Please see our detailed responses below under 'Detailed comments'.

My last point is that there is no discussion on the issue of avoiding causing the emissions during the “peak warming”. Why it is so important?

>> We now added a new paragraph to the introduction of the manuscript, explaining why we compared our RF pathways with peak warming. It reads (L45-49):

“These different atmospheric dynamics are relevant for the question how the various management scenarios will influence global climate and whether a scenario will amplify or attenuate peak global warming, i.e. the maximum deviation in global surface temperatures relative to pre-industrial times. An amplification of peak warming increases the risk of reaching major tipping points in the Earth’s climate system^{13,14}.”

In addition, we also rephrased the Conclusions section of the manuscript to include a discussion of the implications of our findings for peak warming (L109 ff.):

“In conclusion, without rewetting the world’s drained peatlands will continue to emit CO₂, with direct negative effects on the magnitude and timing of global warming. These effects include a higher risk of reaching tipping points in the global climate system and possible cascading effects¹³. In contrast, we show that peatland rewetting can be one measure to attenuate peak global warming: The sooner drained peatlands are rewetted, the better it is for the climate.”

Detailed comments:

line 27: how well these peaks are understood? Do they occur in all types of peatlands? The way the text is written gives a feeling of a lot of uncertainties (possible, may be associated...) even though the peaks have a central role here.

>> The matter of CH₄ peaks upon rewetting was indeed not well addressed in our previous version. CH₄ peaks, higher than the associated CO₂ emission reductions using a GWP approach, have been observed in a handful of rewetted peatlands (in temperate Europe and with prior agricultural use). They have attracted attention of UNFCCC discussions (Couwenberg 2009, Joosten 2009b, Couwenberg & Fritz 2012), the voluntary carbon market (O’Sullivan & Emmer 2011, Verified Carbon Standard 2017) and science (Wilson et al. 2016), but are atypical (Couwenberg & Fritz 2012, Wilson et al 2016).

In our previous version, we used universal CH₄ peaks to illustrate the benefits of rewetting (despite possible CH₄ peaks) under very conservative assumptions. We have now removed the peak from the main graphs and addressed the issue only in a few sentences (L149 ff.) and included an illustrative graph in the supplementary material (Fig. S1).

“Individual studies have discussed the presence of a CH₄ peak for the first years after rewetting^{7,8}. Although this is likely not a global phenomenon²⁴, please see supplementary Figure S1 for an estimate of the uncertainty related to possible CH₄ peaks.”

line 33: may be associated? Isn't this clear? How is this uncertainty reflected in the RF calculations?

>> Please see our previous comments.

lines 50-51: In the rewetting scenarios the reference situation is the emissions from drained peatlands. However, in the “Drain_more” scenario the drained situation cannot be used as a reference. In that sense the scenarios are not balanced. The reference scenario for the drain_more should be given.

>> We use no reference *scenario* but a reference *situation*, which for all scenarios is the same: the current situation in which ~15% of the global peatland area is drained and the rest is undrained. As we conservatively assume that pristine peatlands are climate-neutral (in fact they are small carbon sinks), the scenarios differ only in the areal proportions of drained or rewetted peatland. How pristine peatlands will globally respond to rising global temperatures and changed weather patterns is unclear. Our assumption of climate-neutrality of pristine peatlands allows focusing on drained and rewetted peatlands, i.e. on direct management, causing direct human-induced emissions.

To increase clarity we have now rephrased the text as follows (L60-61):

“For our modeling exercise, we focus on the direct human-induced climatic effects and conservatively assume pristine peatlands to be climate-neutral.”

Note that using a reference *scenario* is not feasible for our analyses: The “No_Change” scenario represents the scenario in which the area of drained peatland is kept constant. We refrained from expressing the other scenarios as differences to this scenario, because the continued CO₂ emissions from a constant area of drained peatland lead to persistently increasing CO₂ emissions in the atmosphere and to intensified climate warming. This counter-intuitive implication (“No_Change” in drained area leads to an increasing climate burden) is a central message of the manuscript. This message would remain obscured by using the “No_Change” scenario as a ‘reference scenario’, as would the negative effect of delaying rewetting (another central point). Meanwhile, using different reference scenarios for the “No_Change” and the other scenarios would unnecessarily complicate the story. Also, using entirely pristine conditions as a reference is impossible because

peatlands in some parts of the world already started to degrade anthropogenically thousands of years ago (e.g. China and Europe, Joosten 2009a), even before major tracks of peatlands started to form and expand elsewhere (e.g. SE Asia, Dommain et al. 2014), including as a result of human activities (e.g. Moore 1993).

line 60: is this valid for “drain_more” only?

>> No, see our comment above.

line 62: I am missing justification for the strong spike in the beginning. Why 10 times the natural? How sensitive your calculation is for this number? What if the size of the spike was 100 times the natural emissions, or less than 10? Would it change your conclusions? Are the results of the only two papers generalizable? Now there is a sensitivity test for the areas of drained/rewetted peatlands, but not for the flux rates, which I suppose are much more critical than the area. Also, where is the peak duration of 0-5 years stemming from?

>> Please see our previous comments.

lines 66-67: Here status quo means that the reference is “No_change”, but it is not explicitly stated. It would be better to present all the results as a difference between the scenario and the reference. Then, also the ‘warming overshoot’ would be more easily and clearly detected.

>> Please see our response above.

I am missing discussion related to the selection and use of climate scenarios. This has not been explained at all and should be done either in the main text or in supplement. Why are you using exactly these scenarios? Why not the 1.5C scenarios which are more recent and show an earlier peak warming? Why peak warming is so important? And which one is more important: the quick warming impact, or the cooling afterwards?

>> In Figure 1, we show all available model/scenario combinations of the IIASA database that give global mean temperatures, including all available 1.5°C scenarios. Please note that scenarios that limit long-term global warming to 1.5°C or 2.0°C often temporarily exceed these temperatures before stabilization. Thus, they may have higher ‘peak temperatures’.

We now better explain the origin of the climate scenarios in the Methods section (L166 ff.) and point out the significance of the relationship to peak warming throughout the text, and rephrased accordingly.

If the RF values in Fig. 1 were shown as cumulative values one could easily see when the warming due to increased CH₄ emissions have been compensated. Also, I find no discussion of the dynamics of the RF curve. Why is the curve for “drain_all” saturating so quickly? Because there will be more CO₂ in the atmosphere and any addition/removal will have smaller relative impact? These explanations and discussion should be provided somewhere.

>> On the topic of showing cumulative values: Please see our answers to reviewer 3.

With respect to the “drain_all” curve, this is probably a confusion: there is no “drain_all” scenario, only two “Rewet_All_xx” scenarios. The reason for the rapid saturation of the “Rewet_All_xx” scenarios is that CO₂ emissions are stopped (almost) entirely and CH₄ concentrations quickly reach a steady-state of decay and emission.

I am also missing more detailed discussion of the RF dynamics and the warming effect: Questions related to the RF calculation which should be addressed somewhere: Why is the temperature change looked at since 2005? Why exactly this year?

>> 2005 is the base year for most emission projections and for projected global temperature change in AR5. Since the starting point for the perturbation model is flexible, we chose the same year for the peatland radiative forcing scenarios to enable direct comparison. We now added clarifying text in the Methods section (L166 ff.):

“We compare the radiative forcing trajectories of the various peatland management scenarios with the global temperature change as projected by all available pathways of IPCC’s AR5²⁰ and use the same starting year 2005 as these pathways.”

You are saying that during the 21st century, peatland drainage will warm the climate by 0.2 K. This is 20% of the already observed global warming. How realistic it is?

>> This figure is of the same order of magnitude as reported by Leifeld et al. 2019, who write “emissions from peatland may comprise 12–41% of the GHG emission budget for keeping global warming below +1.5 to +2 °C without rehabilitation”. Also, Steffen et al. 2018 find that several feedback mechanisms have warming effects in a similar range in 2100 (e.g. permafrost thawing → 0.09 K, weakening of land and ocean physiological sinks → 0.25 K). Nonetheless, thanks to your comment we found and corrected a small error (twisted numbers) in our calculation, so that the warming effect is now slightly smaller.

We added a sentence in L29-31 to set our results in context. It reads:

“A recent study states that the effect of emissions from drained peatlands in the period 2020–2100 may comprise 12–41 % of the remaining GHG emission budget for keeping global warming below +1.5 to +2 °C ⁽²⁾.”

Note that Leifeld et al. 2019 use global warming potentials (also see our answers to Reviewer 2), but as their 12-41% is caused by drained peatlands, in which CO₂ emissions strongly dominate, these values are comparable with our RF results (their rewetting or 'rehabilitation' scenarios less so, as GWP misconstrues the effects of CH₄, Allen et al. 2018).

lines 73-74: the emissions of rewetted peatlands have not been shown anywhere. Please provide them in a table similarly to the emissions of the drained peatlands. Also, N₂O is first time mentioned here. Some description of the processes which lead to the suppressed emissions is necessary.

>> We now included the assigned EF's of rewetted peatlands in Table M1.

With respect to N₂O: we addressed N₂O fluxes from drained peatlands in L40 and L50. N₂O behaves in a similar way as CO₂, but its climatic effects are small in all scenarios (see Figure 2). The main focus of our paper is, however, on the climatic effects of the more important greenhouse gases in this context, CO₂ and CH₄. Therefore, we did not explain the processes that lead to suppressed N₂O emissions in rewetted peatlands and simply refer to the IPCC Wetlands Supplement that provides emission numbers.

line 75: "...quickly reaches a plateau..." But what happens if you reduce the climate impact of the reference here?

>> As the reference is a fixed point in time (2005) and the climate effects of all management scenarios are expressed as a difference compared to that point, the shape of the curves does not depend on the absolute value of the reference point.

lines 75-76: "differences in radiative forcing between the scenarios are mainly determined by CO₂": this is not at all clear from the figure 2. Again, would be more informative to look at the differences between the emissions from /sinks of reference vs. rewetted.

>> Please also see our previous comments regarding reference point vs. scenario. We re-phrased the sentence to make our meaning clearer (L80-82).

"Meanwhile, differences in radiative forcing between the 'drainage' (increased forcing) and 'rewetting' scenarios (stable forcing) are mainly determined by differences in the forcing of CO₂ (Figure 2)."

In Fig. 1c you are stating that it does not include forcing of peatlands that remain pristine. If this means that emissions/sinks of pristine mires have not been used as a reference for the

“drain_more” scenario, it is not OK. For the “drain_more” scenario it is exactly the fluxes of pristine mire which should be used as a reference.

>> Please see our previous answers. The carbon sequestration of pristine mires is an order of magnitude (or more) smaller than the carbon emission from drained peatlands and may therefore be conservatively neglected. Our study does not aim to assess the climatic effects of *all* global peatlands (pristine + drained), but – recalling the aims of the UNFCCC – to focus on the anthropogenic (directly human-induced) emissions/removals: What is the most climate-friendly way to manage peatlands under ‘human control’?

We have now included (L60-61):

“For our modeling exercise, we focus on the direct human-induced climatic effects and conservatively assume pristine peatlands to be climate-neutral.”

lines 80-83: Again, would be good to discuss why is the “peak temperature” so important here?

>> We re-worked the manuscript to clarify the importance of “peak temperature”. Please also see our other comments.

line 82: “total long-term forcing” what is this? Definition?

>> The sentence was indeed poorly phrased. We changed the sentence to (L87-90):

“Comparing the scenarios ‘Rewet_All_Now’ and ‘Rewet_All_Later’ shows that timing of peatland rewetting is not only important in relation to peak temperature, but also with respect to the total accumulated CO₂ and N₂O emissions in the atmosphere and the resulting radiative forcing (Figure 2).”

line 91: “stabilize global climate” is vague terminology. What does stabilization of climate mean?

>> We re-phrased the sentence. It now reads (L101 ff.):

“In order to reach climate-neutrality in 2050 as implied by the Paris Agreement, it is insufficient to focus rewetting efforts on selected peatlands only: to reach the Paris goal, CO₂ emissions from (almost) all drained peatlands have to be stopped by rewetting².”

lines 94-98: I do not really understand this paragraph. How representative are the references used here? At least the other one (Minkinen et al.) is not really a productive forest, but a poorly growing one.

>>The Minkkinen paper is merely used as an example of a paper that proposes to use wood biomass grown on drained peatlands as a substitute for fossil fuels.

I do not understand this justification: “In this study, we did not include this option because wet cultivation methods (‘paludiculture’) could provide similar substitution benefits without CO₂ emissions from drained peat soil.”

>> We have now rephrased this sentence as follows (L106 ff.):

“In this study, we did not include this option because similar biomass-based substitution benefits can also be reached by cultivating biomass on rewetted peatlands²³, i.e. without CO₂ emissions from drained peat soil.”

line 101: why “if”? Do we expect or do we not? This was the main assumption in this MS. Are you not sure about it?

>> The respective sentence has been deleted.

line 102: why it is important to avoid the peak warming?

>> We now include justification for the need to avoid a high peak warming. The respective sections read (L48-49 and L109 ff.):

“An amplification of peak warming increases the risk of reaching major tipping points in the Earth’s climate system^{13,14}.”

“[...] direct negative effects on the magnitude and timing of global warming. These effects include a higher risk of reaching tipping points in the global climate system and possible cascading effects¹³.”

lines 112-113: what are the consequences of this assumption? How sensitive your result is to this?

>> The current distribution of drained peatland is mainly a function of climatic suitability for agriculture (incl. oil palm and short rotation pulpwood), i.e. concentrates on the temperate and (sub)tropical zones. An exception is boreal forestry, especially in Finland and Russia, where draining (and rewetting) has a comparatively small effect on CO₂ fluxes. It can reasonably be expected that future drainage – similarly to the last two decades - will focus more on tropical and subtropical peatlands meaning that our “Drain_More” scenario will underestimate the climatic effects.

We therefore added the sentence (L127 ff.):

“As future drainage – similar to the past two decades¹⁶ - will probably focus on tropical and subtropical peatlands, our ‘Drain_More’ scenario will underestimate the climate effects of future drainage.”

Table M1: fluxes from the rewetted mires are missing

>> We have now included the assigned EF's of rewetted peatlands in Table M1.

lines 120-121: some critical discussion of the IPCC emission factors would be welcomed. What happens with the averaging what you are doing? How reliable are the estimates for the ditch emissions? How realistic it is that the amount of ditches in agricultural peatlands is 5%, even though the number of underground drains seems to be relatively high? The emissions of ditches and DOC export are quite critical for the whole exercise, as they turn the drained peat soils in many regions into sources. Some discussion on that topic would be very important.

>> Drained peat soils are a net source of emissions irrespective of whether ditches and DOC export are taken into account (Wilson et al. 2016). The 5 % ditch area is only an estimate, but was not contested in the review rounds of the IPCC Wetlands Supplement. Only few countries report on ditches in their National Inventory Report and most of them use the default 5% fraction. Canada explicitly mentions 5% ditch area in peat extraction areas; for which Finland mentions 7%. Germany uses a country specific fraction of 1.3%. From Figure 2 it is apparent that CH₄ emissions from drained sites are not decisive in driving the results.

Regarding the determination of uncertainty of the default IPCC emission factors (EF, average values from meta-analysis): one could set up Monte Carlo simulations that draw from the distribution of the EFs (normal distribution for CO₂ and N₂O, log-normal distribution for CH₄). However, as we are looking at *all* peatlands of the world the distribution would be exhausted completely and we would end up with the average value (i.e. EFs) as the best approximation. Therefore, we instead have adapted the sensitivity analyses in the way that we now also show the uncertainty of our results if the emissions varied in the range $EF \pm 10\%$ and $EF \pm 20\%$ (Figure M1). In this way, we cover possible changes of the (average) EFs in future updates of the IPCC Wetlands Supplement that include newly published emission data. As previous updates have shown (Wilson et al. 2016) EFs unlikely change by more than 20 %, since updates still include a large fraction of the studies already included in the previous version.

Finally, we added information explaining our approach to the uncertainty of emission factors (L147 ff.) and one sentence pointing out that CH₄ emissions also occur from ditches in drained peatlands (L85 ff.):

“Values of the emission factors could change slightly when more emission data becomes available. To cover this possibility, we randomly varied all emission factors within a range of 10 % and 20 % uncertainty in our sensitivity analysis (Fig. M1).”

“Note that in the ‘Rewet_Half_Now’ scenario CH₄ forcing is more than half that of the ‘Rewet_All...’ scenarios, because drained peatlands emit CH₄ as well, most notably from drainage ditches.”

Reviewer #2 (Remarks to the Author):

This is an interesting and nuanced paper that is needed on this topic relating to the unintended consequences of rewetting peatlands to sequester carbon. Many of us have learned that there is no free lunch and the process of flooding, which turns off aerobic respiration to favor long term carbon sequestration, also produced conditions that promote methane emissions. Methane is a strong greenhouse gas and with wetlands, the fluxes of methane are very, very strong. My personal concern working on this the topic is the impression some may make that we should not restore wetlands if the methane fluxes are so large. Hence it is warranted to provide a careful analysis to inform policy makers to prevent them from ‘throwing the baby out with the bath water’. Methane is very reactive with OH radical so its lifetime is only about 15 years, compared to CO₂, with a 300 year plus lifetime. So even if methane is produced it will not linger in the atmosphere indefinitely, though it will be converted to CO₂..that is a different story.

I also add that studies like this are important because they are needed to lend nuance to using natural solution to climate change. My one lingering worry is that some policy maker may conclude that restoring wetlands is bad because they also produce methane a stronger greenhouse gas. But we must not throw the proverbial baby out with the bath water. Ecological restoration has many co benefits that need to be considered and weighed, too. Including permanence of the C sink (most upland sinks have a relative short residence time compared to peatlands), protection of land due to sea level rise and storm surges, filtering water, habitat for birds and fish.

Overall, I firmly agree more with the central and salient conclusion of the authors..., without rewetting the world’s drained peatlands will continue to emit CO₂, with direct effects on the magnitude and timing of peak global warming. These CO₂ emissions can effectively be stopped by rewetting. Especially if we expect large CH₄ emission spikes upon rewetting, we should rewet as soon as possible, so that these CH₄ emissions contribute as little as possible to peak warming. Although the CH₄ cost of rewetting may temporarily be substantial, the CO₂ cost of inaction will be much higher.

I like the aspect of the study that raises the question about restoring wetlands now, so the methane effect gets washed out soon enough, as compared to restoring wetlands later. It all makes me think about historical formation of wetlands, preindustrial. If the methane effect did not wash out we may have experiences a bit more warming than we have between glacial and interglacial periods.

>> Thank you very much for your comments. We completely agree with you, and indeed these points were the main motivation for our analyses.

Methods

This paper addresses a so what question in scale by using the global peatland database, rather than focus on a few sites as some of us are guilty of. This is needed to address magnitude and impact at the scale of the atmosphere that processes all of this; I stress this because my one worry with peatland is that they are a super strong source but a small area, so we need to address these issues head on as this paper does.

This work lets the authors show that CH₄ radiative forcing does not undermine the climate change mitigation potential of peatland rewetting. Instead, postponing rewetting increases the long-term warming effect of continued CO₂ emissions. Unlike CO₂ (and N₂O) from drained peatlands that accumulates in the atmosphere, possible CH₄ emission spikes upon rewetting do not add to expected peak warming when rewetting occurs before 2050

Regarding to the computations of greenhouse warming potential I want to make sure they are using the newer Neubauer and Megonigal 2015 Ecosystems Moving beyond global warming potential,.. sustained warming potential method, rather than the older and simpler pulse method. It seems they are using the pulse method. For the sake of doing science best, and in its most defensible manner I strongly urge them to use the better methods.

>> There seems to be a misunderstanding here. We indeed work with sustained fluxes (not pulses). Actually, we are using neither regular (pulse) global warming potential nor sustained global warming potential (Neubauer and Megonigal 2015), because both integrate the warming effect over a fixed time horizon (e.g.100 years). We decided against these approaches, because they fail to emphasize the short-term warming effect of methane and e.g. hide the effect of rewetting timing in relation to peak warming. Instead, we model the radiative forcing of the released GHGs on every individual moment in time using an atmospheric perturbation model (which is basically an updated/enhanced version of the model that Neubauer and Megonigal 2015 based their analyses on).

One of the other reasons I like this work is it helps support wetland development over upland development. The latter are highly sold, but their residence time remains relatively short. I find undecomposed coroms in our peatlands that have been aged 4000 years! Putting carbon in trees or soil of ag or grasslands is bound to return to the atmosphere in less than 100 years in most locations.

My conclusion, is to publish after revision and recompute the analysis with the sustained greenhouse warming potential method. Methane is not emitted as a pulse after rewetting drained lands. It is sustained for many years.

>> You are absolutely right, ecosystem emissions are usually not pulses (although this assumption is still very common, e.g. Leifeld et al. 2019). We indeed assume the methane (and carbon dioxide/nitrous oxide) fluxes to be sustained. Please see our comment above.

Reviewer #3 (Remarks to the Author):

In the manuscript, authors calculate radiative forcing associated with different options for the future management of peatlands and pay particular attention to the seemingly controversial effect of methane. This is an important and timely piece of work although the main finding, namely that despite high methane release peatland rewetting is beneficial with respect to the overall forcing as compared to business as usual is not novel. The different life times and corresponding consequences for calculating radiative forcing over longer time scales has been widely discussed in the literature. The MS merits publication in Nature Communications but authors should pay attention to my major concerns as listed below.

It is not clear whether the exhaustion of peat deposits over time, after drainage, is considered in the calculated scenarios until 2100. Loosing part of former organic soils with drainage will reduce peatland derived CO₂ emissions over time.

>> Yes, this so-called “peat depletion” in case of peatland drainage is factored in. The assumed annual expansion of drained peatland area with 5,000 km² per year is based on recent historical data (Joosten 2017) which include a peat depletion of annually 0.5% of the drained peatland area.

To clarify this factor in the manuscript, we have changed the phrasing “average rate of new peatland drainage” into “average net increase of drained peatland area” in L64.

Figure 1 provides an overview of scenario outcomes. Panel ‘a’ obviously shows various calculations but it remains unclear what they do represent.

>> As a response to reviewer 1 we added a sentence to the Methods section to explain the origin of the global temperature models. Also, we re-worked the manuscript in order to clarify the intentions and significance of our comparison with peak global warming.

In line 88, authors underpin a fact which has long been known (e.g., Frohking and Roulet 2008). There is less novelty in this finding than stated, although the consideration for different scenarios of future peatland use is appreciated.

>> We did not want to claim here that our study is the first to point out the draw-backs of metrics that use fixed time horizons. Rather, we wanted to point out that the importance of rewetting timing can only be illustrated by a radiative forcing model. In our view, this point is important because using GWP (or SGWP) is still so common (e.g. Leifeld et al. 2019). We now rephrased slightly to make our meaning clearer. The section reads (L97 ff.):

“The climate effect is strongly dependent on the concrete point in time that rewetting is implemented. This fact is hitherto insufficiently recognized because it remains hidden by the common use of metrics that involve predetermined time horizons (like GWP or sustained flux variants of GWP).”

In line 91, authors make a political claim that is, in my view, not appropriate for a scientific journal.

>> We think it is important to interpret the significance of our findings in the context of global governance. The conclusions we draw from our “Rewet_Half_Now” scenario are factual, not political, and therefore not inappropriate for a scientific journal in our view. However, the sentence may have been too vaguely written. We have now rephrased the sentence to read (L101 ff.):

“In order to reach climate-neutrality in 2050 as implied by the Paris Agreement, it is insufficient to focus rewetting efforts on selected peatlands only: to reach the Paris goal, CO₂ emissions from (almost) all drained peatlands have to be stopped by rewetting².”

There is particular need to provide much better explanation of methods: Authors are requested to explicitly explain

i) how the climate zones are defined,

>> We used the climate zones as defined by IPCC, as these are also the basis of the IPCC Tier 1 emission factors used in the paper.

ii) where the land-use and management information (deep vs. shallow drained grassland) is coming from.

>> The emission factors come from the IPCC 2014 Wetlands Supplement, the “activity data” (= the areas) from the Global Peatland Database, which includes national data from the most recent UNFCCC National Inventory Submissions and the Nationally Determined Contributions. The text now reads (L118 ff.):

“Drained peatland area was taken from the Global Peatland Database (GPD)¹⁴, which includes *inter alia* national data from the most recent UNFCCC National Inventory Submissions and the Nationally Determined Contributions.”

Table M1 gives an overview of land use classes and assigned EF's of drained but not of rewetted peatlands. The latter should be included.

>> We now included the assigned EF's of rewetted peatlands in Table M1.

Table M1 and line 124. It is unclear what land use class 'agriculture' refers to given that cropland and grassland, which make up agriculture as a whole, are already listed. What does 'agriculture' include and how was this class delineated in the global data base?

>> The class 'agriculture' results from information on peatland utilization that was not differentiated between cropland and grassland in the original data source. The text now reads "... "Agriculture" (i.e. either grassland or cropland when the original data source did not differentiate between these two categories),..." (L122-123)

Line 126: Authors are requested to write out all categories that were lumped and how.

>> We now show the information on the way we aggregated land use classes between the IPCC wetlands supplement and the GPD as supplementary table (Table S1).

In line 126-127 it is stated that EF's for tropical extraction sites equal those of boreal and temperate regions (owing to lack of data), but numbers in Table M1 do deviate from each other with the EF's from tropical being the lowest.

>> Sorry, that was a mistake in our manuscript; the values in the table were actually correct. What we wanted to say was that the IPCC does not give an EF for *methane* from tropical extraction sites; thus we assumed the temperate/boreal EF for methane emissions ($6.1 \text{ kg ha}^{-1} \text{ a}^{-1}$). We changed the text accordingly (L145 ff.).

More details on the radiative forcing model are needed. From the description it seems that authors assume an instantaneous radiative forcing, but see Neubauer 2014 for a comparison between instantaneous and cumulative radiative forcing.

>> Indeed, we used instantaneous radiative forcing (RF) and not cumulative values.

Cumulative RF is used in greenhouse gas *metrics* designed to assess achievement of emission reduction targets. Metrics based on cumulative RF look at the average impact of a gas relative to CO₂ over a fixed time horizon rather than at the actual climate impact at a point in time (see e.g. Caldeira & Myhrvold 2012, Edwards & Trancik 2014, Allen et al. 2018, Balcombe et al. 2018).

As Balcombe et al. (2018) point out about cumulative (=integrated over time) RF: "integrated radiative forcing [...] does not represent the temperature (or other climate) impact". For example, century-scale cumulative RF of an early emission of CH₄ and a late emission of CO₂ can be the same but that the CO₂ emission will result in a much warmer end-of-century climate (Caldeira & Myhrvold 2012). Also Edwards & Trancik (2014) stress this time-dependency of cumulative RF. Instead, they use instantaneous RF as a benchmark to test alternative greenhouse gas metrics.

Meanwhile, climate is driven by instantaneous RF: According to the IPCC, the forced component of the global mean surface temperature (GMST) responds rapidly and almost linearly to effective RF (medium confidence, IPCC AR5 WGI, p.62). Effective RF and instantaneous RF are very closely related for well-mixed GHGs like CO₂, CH₄ and N₂O (see AR5 WGI Ch. 8). For these reasons, direct (i.e. non-cumulative) RF features prominently in IPCC AR5 WGI (see e.g. Figure SPM.5, TS.7 or 8.17).

For these reasons, we show instantaneous RF as an indication for the climate effect, together with a tentative estimate of the effect on global temperature. In addition, we now included the code for the atmospheric perturbation model as supplementary information.

Further, in line 62 it is mentioned that an initial strong CH₄ spike was considered. Does this correspond to the IPCC EF's for rewetted peatlands? Details on the origin and size of that spike are missing. The longevity of the CH₄ spike also remains unclear as well as what authors refer to as 'natural emissions'.

>> In the previous version of the manuscript, we included a CH₄ spike in order to exaggerate the importance of CH₄ and because there are a few studies from Central Europe reporting such an effect (refs 7, 8). In our simulations, we applied that spike somewhat artificially by multiplying the IPCC emission factor for CH₄ from rewetted peatlands by 10 for the first few years. However, as we wrote in the replies to reviewer 1's comments, we now removed that artificial CH₄ spike because it is not sufficiently backed up as a global phenomenon, and only show it in the supplementary Figure S1.

References

Allen MR et al. 2018 A solution to the misrepresentations of CO₂-equivalent emissions of short-lived climate pollutants under ambitious mitigation. *npj Climate and Atmospheric Science* 1(16): 1-8.

Balcombe P, Speirs JF, Brandon N & Hawkes A 2018 Methane emissions: choosing the right climate metric and time horizon. *Environmental Science: Processes & Impacts* 20: 1323–1339.

Bonn A, Allott T, Evans M, Joosten H & Stoneman R (eds.) 2016 Peatland restoration and ecosystem services: Science, policy and practice. Cambridge University Press/ British Ecological Society, Cambridge, 493 p.

Caldeira K & Myhrvold NP 2012 Temperature change vs. cumulative radiative forcing as metrics for evaluating climate consequences of energy system choices. *Proceedings of the National Academy of Sciences of the United States of America* 109(27): E1813.

- Couwenberg J 2009 Methane emissions from peat soils (organic soils, histosols). Facts, MRV-ability, emission factors. Wetlands International, Ede, produced for the UN-FCCC meetings in Bonn, August 2009, 16 p.
- Couwenberg J & Fritz C 2012 Towards developing IPCC methane 'emission factors' for peatlands (organic soils). *Mires and Peat* 10(3): 1–17.
- Dommain R, Couwenberg J, Glaser PH, Joosten H & Suryadiputra INN 2014 Carbon storage and release in Indonesian peatlands since the last deglaciation. *Quaternary Science Reviews* 97: 1-32.
- Dommain R, Froelking S, Jeltsch-Thömmes A, Joos F, Couwenberg J & Glaser P 2018 A radiative forcing analysis of tropical peatlands before and after their conversion to agricultural plantations. *Global Change Biology* 24: 5518-5533.
- Edwards M & Trancik J 2014 Climate impacts of energy technologies depend on emissions timing. *Nature Climate Change* 4: 347–352.
- Joosten H 2009a Human Impacts: Farming, Fire, Forestry and Fuel. In: Maltby E & Barker T (eds.), *The Wetlands Handbook*. Blackwell Publishing, pp. 689-718.
- Joosten H 2009b The long and winding peatland road to Copenhagen, stage Bonn III. *IMCG Newsletter* 2009-2: 20-23.
- Joosten H 2017 The development of peatland emissions until 2030: a reconnaissance. *IMCG Bulletin* 9: 4-8.
- Leifeld J et al. 2019 Intact and managed peatland soils as a source and sink of GHGs from 1850 to 2100. *Nature Climate Change*, doi: 10.1038/s41558-019-0615-5
- Moore PD 1993 The origin of blanket mire, revisited. In: Chambers F.M. (ed.), *Climate Change and Human Impact on the Landscape*. Chapman and Hall, London, pp. 217–224.
- Neubauer SC & Megonigal JP 2015 Moving Beyond Global Warming Potentials to Quantify the Climatic Role of Ecosystems. *Ecosystems* 18: 1000-1013.
- O'Sullivan R & Emmer I 2011 Selling peatland rewetting on the voluntary carbon market. In: Tanneberger F & Wichtmann W (eds.) *Carbon credits from peatland rewetting. Climate - biodiversity - land use. Science, policy, implementation and recommendations of a pilot project in Belarus*. Schweizerbart, Stuttgart, pp. 94-99.
- Steffen W et al. 2018 Trajectories of the Earth system in the anthropocene. *Proceedings of the National Academy of Sciences of the United States of America* 115(33): 8252–8259.
- Verified Carbon Standard 2017 VM0036 Methodology for Rewetting Drained Temperate Peatlands v1.0, <https://verra.org/methodology/vm0036-methodology-for-rewetting-drained-temperate-peatlands-v1-0/>

Wichtmann W, Schröder C & Joosten H (eds.) 2016 Paludiculture – productive use of wet peatlands. Climate protection – biodiversity – regional economic benefits. Schweizerbart Science Publishers, Stuttgart, 272 p.

Wilson D, Blain D, Couwenberg J, Evans CD, Murdiyarso D, Page SE, Renou-Wilson F, Rieley JO, Sirin A, Strack M & Tuittila E-S 2016 Greenhouse gas emission factors associated with rewetting of organic soils. *Mires and Peat* 17(4): 1–28.

REVIEWERS' COMMENTS:

Reviewer #1 (Remarks to the Author):

I have now read the response of the authors to my reviewer comments given earlier and found that they have responded adequately to all comments. On my behalf, I can recommend publication of the manuscript in Nat. Comm.

Reviewer #3 (Remarks to the Author):

It was a pleasure reading the rebuttal letter; the authors addressed all comments by the reviewers appropriately. The text also has become clearer and easier to follow. Provision of the perturbation model underpinned the validity of the work. I suggest acceptance of the work after consideration of the points listed below.

Abstract. The last sentence is not correct; there is of course an effect of CH₄ from rewetting (as the study shows), but it plays a minor role for the overall radiative effect of peatland rewetting. I suggest to delete the last sentence as it is mostly a political statement ('counterproductive').

Line 64. The 'average net increase' accounts for the disappearance of drained peatlands that fade away after complete loss of their peat, but this is known only to the reviewers of the authors MS and, hence, readers of their rebuttal letter. It should be clearly stated in the methods section.

Line 101. The Paris Agreement does not imply 'climate neutrality by 2050'; this implication comes from scientific studies and interpretations carried out thereafter/elsewhere. It clearly sets a temperature goal ('Holding the increase in the global average temperature to well below 2°C above pre-industrial levels and pursuing efforts to limit the temperature increase to 1.5°C above pre-industrial levels') and this is what should be referred to here.

Line 149 and response to first reviewer. It should become clear from the main text that the chosen EF uncertainties are arbitrary and smaller than those reported in the IPCC wetland supplement. The approach itself is reasonable, but should not lead to confuse anticipated with measured EF uncertainties.

REVIEWERS' COMMENTS:

Reviewer #1 (Remarks to the Author):

I have now read the response of the authors to my reviewer comments given earlier and found that they have responded adequately to all comments. On my behalf, I can recommend publication of the manuscript in Nat. Comm.

Reviewer #3 (Remarks to the Author):

It was a pleasure reading the rebuttal letter; the authors addressed all comments by the reviewers appropriately. The text also has become clearer and easier to follow. Provision of the perturbation model underpinned the validity of the work. I suggest acceptance of the work after consideration of the points listed below.

Abstract. The last sentence is not correct; there is of course an effect of CH₄ from rewetting (as the study shows), but it plays a minor role for the overall radiative effect of peatland rewetting. I suggest to delete the last sentence as it is mostly a political statement ('counterproductive').

>> We deleted the respective sentence.

Line 64. The 'average net increase' accounts for the disappearance of drained peatlands that fade away after complete loss of their peat, but this is known only to the reviewers of the authors MS and, hence, readers of their rebuttal letter. It should be clearly stated in the methods section.

>> We added a clear statement on this topic in the Methods section (L124 ff.).

Line 101. The Paris Agreement does not imply 'climate neutrality by 2050'; this implication comes from scientific studies and interpretations carried out thereafter/elsewhere. It clearly sets a temperature goal ('Holding the increase in the global average temperature to well below 2°C above pre-industrial levels and pursuing efforts to limit the temperature increase to 1.5°C above pre-industrial levels') and this is what should be referred to here.

>> We changed the sentence to:

"Finally, in order to reach climate-neutrality in 2050 (as implied by the Paris Agreement on limiting the increase in global temperature to well below 2 °C), it is insufficient to focus rewetting efforts on selected peatlands only: to reach the Paris goal, CO₂ emissions from (almost) all drained peatlands have to be stopped by rewetting²" (L99 ff.)

Line 149 and response to first reviewer. It should become clear from the main text that the chosen EF uncertainties are arbitrary and smaller than those reported in the IPCC wetland supplement. The approach itself is reasonable, but should not lead to confuse anticipated with measured EF uncertainties.

>> We now added an explanation of our choice for the sensitivity analyses in L147 ff.:

“Please note that these uncertainty ranges do not correspond to the confidence intervals given by the IPCC Wetlands Supplement, since those describe the observed variability of emissions from individual peatlands. Since our analyses take a global perspective, our sensitivity analyses instead cover possible changes of the mean emissions (i.e. emission factors).”